# Origin of agricultural plant pathogens: Diversity and pathogenicity of *Rhizoctonia* fungi associated with native prairie grasses in the Sandhills of Nebraska

Srikanth Kodati[1,2☯¤a], Anthony O. Adesemoye[1,2☯¤b], Gary Y. Yuen[1☯], Jerry D. Volesky[2☯], Sydney E. Everhart [1☯]*

1 Department of Plant Pathology, University of Nebraska, Lincoln, Nebraska, United States of America,
2 West Central Research and Extension Center, University of Nebraska, North Platte, Nebraska, United States of America

☯ These authors contributed equally to this work.
¤a Current address: The Connecticut Agricultural Experiment Station Valley Laboratory, Windsor, Connecticut, United States of America
¤b Current address: Terramera Inc., Vancouver, British Columbia, Canada
* everhart@unl.edu

**Data Availability Statement:** UNL Data Repository. Dataset. doi: 10.32873/unl.dr. 20200825.

## Abstract

The Sandhills of Nebraska is a complex ecosystem, covering 50,000 km$^2$ in central and western Nebraska and predominantly of virgin grassland. Grasslands are the most widespread vegetation in the U.S. and once dominated regions are currently cultivated croplands, so it stands to reason that some of the current plant pathogens of cultivated crops originated from grasslands, particularly soilborne plant pathogens. The anamorphic genus *Rhizoctonia* includes genetically diverse organisms that are known to be necrotrophic fungal pathogens, saprophytes, mycorrhiza of orchids, and biocontrol agents. This study aimed to evaluate the diversity of *Rhizoctonia* spp. on four native grasses in the Sandhills of Nebraska and determine pathogenicity to native grasses and soybean. In 2016 and 2017, a total of 84 samples were collected from 11 sites in the Sandhills, located in eight counties of Nebraska. The samples included soil and symptomatic roots from the four dominant native grasses: sand bluestem, little bluestem, prairie sandreed, and needle-and-thread. Obtained were 17 *Rhizoctonia*-like isolates identified, including five isolates of binucleate *Rhizoctonia* AG-F; two isolates each from binucleate *Rhizoctonia* AG-B, AG-C, and AG-K, *Rhizoctonia solani* AGs: AG-3, and AG-4; one isolate of binucleate *Rhizoctonia* AG-L, and one isolate of *R. zeae*. Disease severity was assessed for representative isolates of each AG in a greenhouse assay using sand bluestem, needle-and-thread, and soybean; prairie sandreed and little bluestem were unable to germinate under artificial conditions. On native grasses, all but two isolates were either mildly aggressive (causing 5–21% disease severity) or aggressive (21–35% disease severity). Among those, three isolates were cross-pathogenic on soybean, with *R. solani* AG-4 shown to be highly aggressive (86% disease severity). Thus, it is presumed that *Rhizoctonia* spp. are native to the sandhills grasslands and an emerging pathogen of crops cultivated may have survived in the soil and originate from grasslands.

**Funding:** Financial support of this research was provided, in part, by the University of Nebraska Agriculture Research Division of the Institute of Agriculture and Natural Resources and the USDA National Institute of Food and Agriculture (NEB-28-112) to S.E.

**Competing interests:** Author AA is employed by Terramera Inc. However, this affiliation arose after the conclusion of this study. There are no patents, products in development or marketed products to declare. This does not alter our adherence to PLOS ONE policies on sharing data and materials.

## Introduction

Natural ecosystems, such as grasslands, contain a higher level of genetic and environmental heterogeneity compared to the agroecosystems comprised of one or a few plant species with extreme genetic uniformity [1]. It would therefore be expected to have a high diversity of microorganisms associated with those plants in natural ecosystems. Conversion to agricultural croplands represents a dramatic form of natural ecosystem change. This can lead to selection for and emergence of plant pathogens that possess pathogenicity, the ability to cause disease, on cultivated crop plants. In terms of plant pathogens, it is expected that the high diversity of plant species in a natural ecosystem favors low virulence, generalist pathogens that can infect many host plants [2]. For example, a negative correlation was reported between biodiversity and disease pressure, demonstrating that host diversity also reduced the effect of disease [3, 4]. In contrast, a monocultured agroecosystem provides a homogeneous environment with low plant species diversity at high density that would necessarily reduce the diversity of plant pathogens. Natural environments that are converted to monoculture would therefore represent a selection process on the pathogen populations. It is expected that this bottleneck process would select for pathogens of the crop host plant and, over time, will select for more virulent/ specialized pathogens in that ecosystem [5]. The ability of the pathogen to survive and acclimatize to the new agroecosystem from the natural ecosystem, however, is highly dependent on its evolutionary potential and life history traits [6].

Among all-natural ecosystems, grasslands are the most widespread vegetation in the U.S. [7], and so it stands to reason that some of the current plant pathogens of cultivated crops originated from grasslands, and that surveys to catalogue plant pathogens, particularly soilborne plant pathogens, would yield fungal isolates capable of causing disease on cultivated crops. Between 2006 and 2011, 530,000 ha of grasslands were converted to corn/soybean cultivation throughout a region called the western corn belt, which includes North Dakota, South Dakota, Nebraska, Minnesota, and Iowa [8]. This conversion is thought to be driven by an increase in the price for corn and soybean for biofuel production [9]. Interestingly, this conversion was higher in the western side of the states where the evapotranspiration rate is higher than the mean annual precipitation [10], and is therefore likely due to an increasing number of center pivot irrigation systems. For example, because of a large supply of irrigation water for agriculture in Sandhills of Nebraska, this region has been in different states of transition into cultivated land over the past 100 years. Despite these transitions occurring over past century, little information about the diversity of native soilborne fungi and whether pathogens on cultivated crops may have originated from the native grasslands.

The semi-arid Sandhills region of Nebraska contains large sand masses that were formed by blowing sand between 8,000 and 5,000 years ago, and are stabilized by the vegetation, consisting mainly of grasses [11]. Despite limited precipitation and a higher rate of evapotranspiration, soil in the Sandhills are permeable and covered with grasses, which increases infiltration and recharges the U.S. High Plains Aquifer [12]. Water in this aquifer is credited with supporting more than 25% of the nation's agricultural production [13] and spreads underneath 441,160 km$^2$ of land from South Dakota to Texas [14]. The majority of cropland in Nebraska is irrigated using well water, and there are almost 107,000 registered irrigation wells in the state [15]. Nebraska is highest in overall irrigated area (approximately, 3.5M ha) among the 50 states [16]. Cultivation of cool-season grasses, such as smooth brome, and legumes, such as alfalfa and vetch, were some of the first cultivated crops introduced primarily for grasing cattle [17]. A wide variety of other cultivated crops have been grown, primarily in the fringes of the Sandhills region, including (in descending order of area) hay, corn, wheat, dry bean, sugarbeet, sunflower, potato for seed tuber production, and,

beginning in the 1990's, soybean [18]. These attempts to plant row crops have waxed and waned over the years with changes in irrigation and production costs relative to market value. As of 2012, nearly every county in the Sandhills region had over 50% of land area in "farms" [18]. With these cultivated crops also comes the possibility for introduction of other dicots, such as leafy spurge, via seed contamination.

The large diversity of plant species in grasslands might harbor diverse groups of soilborne fungi [19], many of which colonize plant roots [20]. Therefore, it is hypothesized that the heterogeneous natural plant ecosystem of the Sandhills grassland may serve as a continuous source of pathogens affecting neighboring crops or survive in the soil to colonize cultivated crop plants once it is converted to homogeneous agroecosystem. Some pathogens may even become pathogens of significance on cultivated crop plants. In one of the studies conducted on plant-fungal diversity relationships in a grassland ecosystem, a significant number of operational taxonomical units were found belonging to the same family of basidiomycete fungi in the genus *Rhizoctonia* [21]. The genus *Rhizoctonia* is a species complex with several species that are considered cosmopolitan, some of which are known to be necrotrophic plant pathogens that are soilborne and cause mainly seedling diseases on a wide range of plant species. This group also includes saprophytes that live on dead plant tissue and organic matter and are known to form mycorrhizal associations with orchids [22]. Several species of *Rhizoctonia* have been reported to cause disease on native and introduced grass species found in the Sandhills. The most widely described species, *R. solani*, is reported as a pathogen of 158 species in the Poaceae [23], 22 of which are catalogued as native and 25 as introduced to the Sandhills [17]. Each of four other *Rhizoctonia* species, *R. cerealis*, *R. oryzae*, *R. zeae*, and *R. practicola*, are reported as pathogens on up to 13 grass species that can be found in the Sandhills, most being non-native; among the native grass species, *Panicum virgatum* is a host for *R. cerealis* and *R. practicola*, while *Schizachyrium scoparium* is a host for *R. oryzae* [17, 23].

Among *Rhizoctonia* species reported from grasses known in the Sandhills, several are pathogens of cultivated grasses, including *R. solani*, *R. cerealis*, *R. oryzae*, and *R. zeae*. For example, *R. solani* causes brown patch disease in cool season turf grasses (*Agrostis*, *Festuca*, *Poa*, and *Lollium*) and large patch disease in warm season turf grasses (*Buchloe*, *Cynodon*, *Stenotaphrum*, and *Zoysia*) [24]. Within that species, anastomosis groups (AG) are used to define different groups that can have host specificity and it is mostly AG-1, 2–2, and 4 and occasionally AG-5 and -6 that cause disease on turfgrasses [24]. The pathogen *R. cerealis* (AG-D) causes yellow patch disease in the same species as *R. solani*, while *R. oryzae* and *R. zeae* cause leaf spot and sheath blight on *Agrostis*, *Eremochloa*, and *Stenotaphrum* [24]. No previous studies have reported *Rhizoctonia* spp. on grasses or soils of the Sandhills in Nebraska and their ability to cause disease on native grasses and cultivated crops [25].

There is a need to better understand the diversity of soilborne pathogens, specifically the *Rhizoctonia* spp. in the native diversity within the Sandhills, with grasses being the predominant members of that plant community. One objective of this study is to determine the *Rhizoctonia* species that occur among the grass species, and the biological relationships that those organisms have with those plants. Because of the continued transition of these regions due to anthropogenic disturbances, this region is a dynamic zone, therefore, the second objective of this study is to assess the potential for *Rhizoctonia* spp. from grasses to infect soybean, with soybean as a model, commonly cultivated crop in Nebraska. This information is useful for increasing our knowledge of the potential for native plants to harbor fungal plant pathogens that can cause diseases on cultivated crops, and to identify fungal pathogens that may be potential threats as additional areas within the Sandhills grassland are converted to crop cultivation.

## Materials and methods

### Sandhills region

The Sandhills region is defined as an area of approximately 50,000 km$^2$ in central and western Nebraska, with variation in precipitation and in the height and lengths of sand dunes. The annual precipitation is highest in the east, receiving 65 cm at an average elevation of 550 m, and 55 cm at an average elevation of 1,100 m in the west [12]. The mean temperature is 9.4˚C in the eastern Sandhills and less than 8.9˚C in the western portion [26]. The Sandhills is primarily tallgrass prairie with approximately 720 species of plants and 670 (93%) of those are native species. Grass species include C$_3$ (cool season) and C$_4$ (warm season) grasses [27], which can be further classified as being either tall or short grass species. In Nebraska, cool season grasses start actively growing in April and continue until cool temperatures and rains prevail, typically November. Cool season grass species do not go dormant in the winter and instead go into a metabolically inactive state, surviving freezing temperatures by formation of complex sugars that act as antifreeze agents [28]. The most common cool season grasses are needle-and-thread (*Hesperostipa comata* Trin. & Rupr.), and porcupine grass (*Hesperostipa spartea* Trin.) [29]. Common warm season grasses in the Sandhills are little bluestem (*Schizachyrium scoparium* [Michx.]), switchgrass (*Panicum virgatum*), prairie sandreed (*Calamovilfa longifolia* [Hook] Scribn.), and sand bluestem (*Andropogon hallii* Hack).

### Sample collection

Plants with soil were collected in the month of June in 2016 and 2017 from 11 sites within the Sandhills that had never been cultivated but may have been grazed, which were on private and public lands in eight counties in Nebraska: Arthur, Brown, Grant, Hooker, Thomas, Keith, Logan, and Rock (Fig 1). Four species of grasses were collected from each location: sand bluestem, little bluestem, prairie sandreed, and needle-and-thread (Fig 2). At each location, patches or clumps of a single grass species were sampled to minimize the number of roots of other plant species. Plants and root zone soil were collected by carefully digging up a clump of plants with a shovel, attempting to minimize root damage and placed into a one-gallon (3.79 dm$^3$) size plastic zipper storage bag. The amount of soil collected into each plastic bag was approximately 1 dm$^3$. Four such samples were collected from each location, one for each of the four native plant species. After collecting each sample, the shovel was sterilized by spraying with alcohol (75% ethanol) followed by flaming using a handheld utility gas lighter. Once excavated, plants and soil were placed into gallon size plastic bags (3.79 dm$^3$), put on ice in a cooler and transported to the laboratory. Samples were stored at 4˚C until processing, and not stored

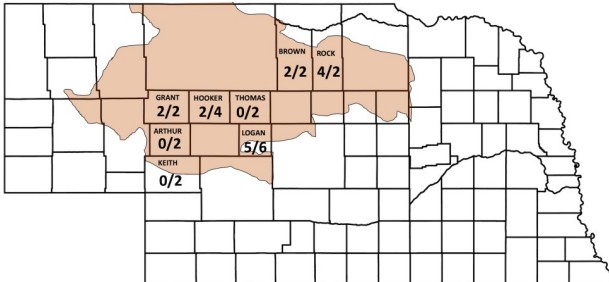

**Fig 1. Map of the Sandhills.** Shaded in red are the Sandhills in Nebraska, where grass and root zone soil were collected in 2016 and 2017 for the present study. *Rhizoctonia* spp. were isolated from samples collected from 11 sites in Brown, Grant, Hooker, Logan, and Rock Counties of total eight counties. These sites were selected because they had never been cultivated but may have been grazed. Numbers in each county indicate the isolates/field-years.

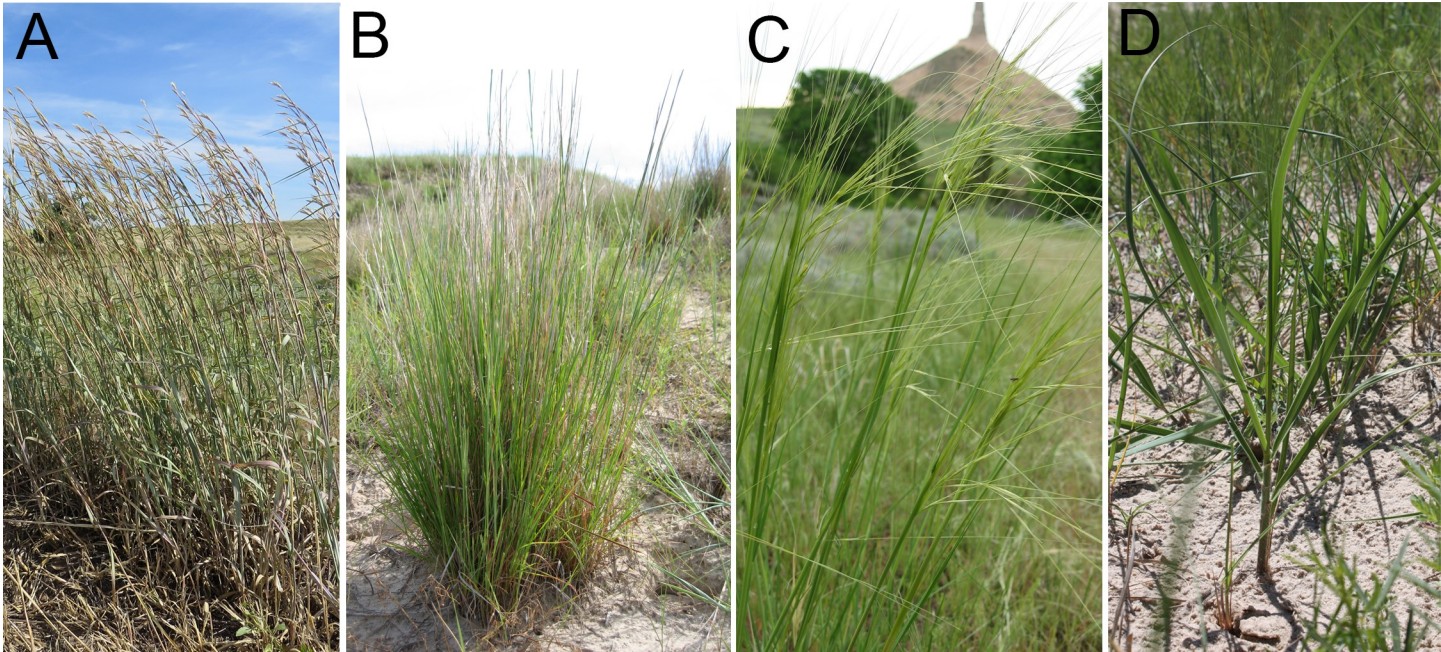

**Fig 2. Grasses surveyed in the study.** Shown are: A) sand bluestem (*Andropogon hallii* Hack), B) little bluestem (*Schizachyrium scoparium*), C) needle-and-thread (*Hesperostipa comata*), and D) prairie sandreed (*Calamovilfa longifolia*).

longer than two days. Among the 11 locations sampled in 2016, and ten locations in seven counties were resampled in 2017. In total, 84 samples (plants and soil) were processed in two years, which was 21 samples associated with each of the four native grass species.

## Isolation of *Rhizoctonia* spp. from plants

Different procedures were used to isolate *Rhizoctonia* from plants and from soil samples. Plants were separated from the soil by gently shaking plants and retaining soil to be processed separately. Roots of the plants were washed under slow-running, room temperature tap water to remove rhizosphere soil and debris. Two symptomatic roots from each plant were selected and cut into pieces of 3–5 mm² with a pre-sterilized scalpel. Root pieces were disinfested in 75% ethanol for 30 s and then placed in sterile deionized water for 1 min. Excess water on root pieces was removed by placing them on a wooden plank that was first surface disinfested by wiping with 75% ethanol and allowed to dry. The root pieces were aseptically transferred onto Petri plates containing Rhizoctonia selective medium (RSM), a semi-selective medium that allows *Rhizoctonia* spp. colonies to grow and hinders or slows the growth of most other fungi and bacteria. The medium consists of 18 g agar, 100 mg streptomycin sulfate, 100 mg penicillin-G sodium salt, and 800 µl 1M sodium hydroxide in 1 L sterile deionized water. RSM plates containing roots were incubated at 22 ± 1°C for 48 h.

## Isolation of *Rhizoctonia* spp. from soils

Root zone soil was baited for *Rhizoctonia* spp. as follows: soil from each sample was placed in a pre-sterilized pot (15 cm diameter and 11 cm deep) and mixed with water to 15% wt./wt. Four 5 cm long sterile wooden birch toothpicks were inserted into the soil, leaving one-fourth of its length above the soil line. The soils with toothpicks were incubated on tables in the greenhouse

at 21 ± 2˚C for 48 h. Toothpicks were collected from each pot and individually placed onto RSM media in Petri plates, which were incubated for 36–48 h at 22 ± 1˚C.

After the incubation period, plates containing roots or toothpick baits were examined under a stereo microscope at 400× magnification (Micromaster, Fisher Scientific) for right-angled branched hyphae characteristic of *Rhizoctonia*. Preliminary identifications of *Rhizoctonia* spp. were made using morphological features, which include right angle branched hyphae branching near the distal septum, a constriction in the hypha at the branch point of origin, and formation of septa near the branch point of origin. Hyphal tips from the margin of *Rhizoctonia* colonies were aseptically transferred onto potato dextrose agar (Difco Laboratories, Detroit, MI) amended with 0.01% tetracycline (PDAt), and water agar and incubated for 5–7 days. All cultures were serially transferred to fresh PDAt plates until pure cultures were obtained. Not more than two *Rhizoctonia* spp. isolates from the roots of a plant sample and one isolate from the associated soil were retained for species and AG identification.

## Classification and identification of *Rhizoctonia*

One of the characteristics used for classification in the genus *Rhizoctonia* is the number of nuclei present in a young hypha. The majority of species recognized as plant pathogens are classified on this basis as either binucleate *Rhizoctonia* (teleomorph: *Ceratobasidium* spp.) or multinucleate *Rhizoctonia* [30], which includes several species such as *R. solani* (teleomorph: *Thanatephorus cucumeris*) and *R. zeae* (teleomorph: *Waitea circinata* var. *zeae*). The *Rhizoctonia* species complex is further divided into different anastomosis groups (AGs) based on their inability to anastomose hyphae between members of different groups. Due to the complex nature of these organisms, identifications to AG is performed by sequencing the ITS and β-tubulin genes and performing a BLAST search of GenBank.

Approximately 2 cm$^2$ of hyphal fragments were harvested from five-day-old isolates of *Rhizoctonia* spp. and added to a 2 ml screwcap extraction tube. DNA was extracted using a buffer extraction method that was described previously [31], following which purified DNA was stored at -20˚C until further use. Polymerase chain reaction (PCR) for amplification of ITS1, ITS2, and 5.8S regions of ribosomal DNA (rDNA) was performed using the following primer set: Internal transcribed spacer (ITS) forward primer 4 (5′-TCCTCCGCTTATTGATATGC-3′) and ITS 5 reverse primer (5′-GGAAGTAAAAGTCGTAACAAGG-3′; [32]). Primers used to amplify the β-tubulin gene regions were: B36F (5′-CACCCACTCCCTCGGTGGTG-3′) and B12R (5′-CAT GAAGAAGTGAAGACGCGGGAA-3′; [33]. PCR was performed in a total volume of 25 µl reaction, which was comprised of 12.5 µl of GoTaq Green Master Mix (Promega Corporation, Madison, WI), 0.6 µl of each of 10µM forward (ITS4) and reverse (ITS5) primers, 4 µl of genomic DNA, and 7.3 µl nuclease free water. No modifications were made in thermal cycling parameters for ITS [34] and B36F/12R primers [33]. To confirm PCR yielded the expected amplicon size, PCR products were separated using electrophoresis on 1% agarose gel in 1 X Tris-boric acid-EDTA (TBE) buffer and visualized using a Molecular Imager® Gel Doc XR + Imaging System 1 (Bio-Rad Laboratories, Inc., Hercules, CA). Fragment sizes were estimated by comparing to bands in a 100 bp ladder stained with SYBR green nucleic acid dye.

## Amplicon sequencing and characterization

PCR products were purified using a QIAquick PCR Purification kit (QIAGEN© Inc., Valencia, CA), and the quantity and quality of amplicons measured using a spectrophotometer (Nano-Vue Plus, GE Healthcare Life Sciences, Marlborough, MA, U.S.). Sanger sequencing was conducted on the ABI 3730XL sequencer at the Institute of Integrative Genome Biology, University of California, Riverside. The Seqman Pro, Seqbuilder, and EditSeq modules of the

DNASTAR Lasergene software v15 (DNASTAR, Madison, WI) were used to edit and align sequences. Sequences were then compared with sequences from National Center for Biotechnology Information (NCBI) database using Nucleotide Basic Local Alignment Search Tool (BLASTn) (http://www.ncbi.nlm.nih.gov/). The database sequence to which each query sequence obtained a match of more than 98% similarity was considered a matching identity. For any query sequence with multiple matches above 98% similarity, the highest matching score was considered the matching identity. Consensus sequences will be submitted to Gen-Bank. Based on the sequence of the ITS and β-tubulin gene regions, the AG of isolates could be identified.

## Soybean pathogenicity bioassay

Because of the difficulties in germinating seeds of the native grass species, it was not possible to assess the pathogenicity of every isolate from a plant directly on the plant species from which the isolate originated. Therefore, a rolled towel assay using soybean as the host plant was used to screen all isolates for potential plant pathogenicity and reduce the number of isolates for further evaluation in greenhouse assays. Seeds of soybean (CR2603, donated by Monsanto Company) were first disinfested by soaking in 2% solution of commercial bleach (5.25% sodium hypochlorite, Clorox Company, Oakland, CA) for two min and rinsed with sterile deionized water for two min each, repeated for a total of three times. Then, for each isolate, eight seeds were arranged evenly in a row on a moistened sterile paper towel and each seed was inoculated with 100 µl of mycelial suspension of a *Rhizoctonia* isolate. Mycelial suspensions were prepared for each isolate separately by scraping the mycelium from a 7-day-old culture of the isolate in a Petri plate with a pre-sterilized scalpel into 10 ml of sterile deionized water. Following inoculation, the seeds were covered with a second layer of moistened sterile paper towel, and the paper towels were rolled together and placed in a 2 L pitcher. Three such rolled-towel units were prepared for each isolate, which then were kept in a randomized array in a reach-in plant growth chamber (E-41L2; Geneva Scientific LLC, Fontana, WI) for 7 days at 21 ± 1°C, 75% relative humidity, with 12 h of photoperiod. Rolled towels without fungal inoculum were prepared as negative controls. This experiment was repeated.

Rolled towels were opened after the incubation period and each seedling was rated individually for disease severity using a rating scale for soybean previously described [34, 35]. The nine-point rating scale for soybean was as follows: 1) no visible symptoms and normal plant development, 2) small, superficial root lesions and normal plant development with 1–10% of the root system of seedling being symptomatic, 3) small superficial root lesions and normal plant development with 11–20% symptomatic, 4) deep advanced root/hypocotyl lesions and normal plant development with 21–35% symptomatic, 5) deep more advanced root/hypocotyl lesions and reduced plant and secondary root development with 36–50% symptomatic, 6) deep more advanced root/hypocotyl lesions and highly reduced plant and secondary root development with few roots visible and 51–65% symptomatic, 7) deeper more advanced root/hypocotyl lesions and plant and secondary root development highly reduced with barely any or no roots visible, no formation of trifoliate leaves, and 66–80% symptomatic, 8) seedling barely emerged but no substantial growth, absence of secondary roots, and 81–95% symptomatic, 9) seed dead, no emergence, no root, and more than 95% lesion coverage of the seed. The rating from each seedling was converted to the midpoint of the percent disease severity within each 1 to 9 rating, i.e. 0, 5, 15, 28, 43, 58, 73, 88, and 98%, respectively. Average percent disease severity among the eight seedlings from each replicate then was used in the statistical analysis. Isolates were categorized as pathogenic when there was a significant difference from the negative control at the 95% confidence level.

## Aggressiveness on soybean and native grasses

One isolate was selected from each *Rhizoctonia* species or AG for was evaluated for pathogenicity and aggressiveness on grasses and soybean in the greenhouse. At least one isolate from each group with high disease severity in the soybean rolled towel assay was selected. In total, nine *Rhizoctonia* isolates were tested in the greenhouse experiments. Only sand bluestem and needle-and-thread grasses were tested as potential hosts of these isolates because little bluestem and prairie sandreed seeds failed to germinate. Experiments using soybean, sand bluestem, and needle-and-thread were conducted separately, but simultaneously, and each experiment was conducted twice.

For inoculations of soybean, an individual soybean seed (CR2603) was placed into a cone-tainer (SC10 Super, 3.8 cm diameter and 21 cm depth) that was three-quarters filled with potting mix. Two agar plugs from a 7-day-old culture of a *Rhizoctonia* isolate were placed on either side of the seed, which was then covered with potting mix.

Since seeds of sand bluestem and needle-and-thread (Stock seed farms, Inc. Murdock, NE) failed to germinate when planted directly into the potting mix in cone-tainers, two different methods were used to germinate these seeds. In the first method that was used for soybeans, seeds were placed between two layers of sterile moistened paper towels, rolled up, placed on end in a pitcher (2 L), and incubated for 15 days inside a reach-in growth chamber (E-41L2; Geneva Scientific LLC, Fontana, WI). In the second method that was used for the germination of grass seeds, a thermoformed germination tray (27.94 cm W X 54.28 cm L X 6.2 cm D) was filled with a layer of general purpose soilless potting mix (Promix®, Premier Horticulture Inc., Canada). Approximately one hundred seeds of each grass species were evenly spread and covered with another layer of potting mix and incubated in a growth chamber for 15 days at $25 \pm 1°C$. Trays with seeds were moistened by gently sprinkling with tap water once every other day. To inoculate the native grasses, one 15-day-old grass seedling was transplanted to a cone-tainer that was three-quarters filled with potting mix, and two agar plugs of a 7-day-old *Rhizoctonia* isolate placed on either side of the seedling, which was then covered with potting mix.

The negative control treatments in all experiments were PDAt plugs without fungal mycelia that came from non-inoculated plates. Treatments were arranged in a randomized complete block design with four replicates for each treatment. Cone-tainers were overhead irrigated for 15 min twice daily with an automated sprinkler irrigation system, to deliver of approximately 25 ml of water per cone-tainer per day. Plants were allowed to grow for 21 days in the greenhouse at $21 \pm 2°C$ without any nutritional supplement. After 21 days, plants were harvested and rated for disease severity (Fig 3). Soybean seedlings were rated for disease severity on individual seedling using the 1 to 9 rating scale applied in the soybean rolled towel assay and the percent disease severity calculated from the ratings as described above. Grass seedlings were rated according to a 1 to 8 scale adapted from that for corn and wheat [34]. Each rating was converted percent disease severity based on midpoints of percent disease severity in each of the 1 to 8 ratings: 0, 5, 15, 30, 50, 73, 90, and 98%, respectively.

## Data analysis

Data from each plant species in the greenhouse experiments were analyzed separately. Isolates were compared for disease severity using PROC MIXED (SAS version 9.4, SAS Institute, Cary, NC), with data from both experiments with a plant species being analyzed together [36]. Treatment means were separated using Fisher's LSD test with alpha set to 0.05 and adjusted with the Bonferroni correction for multiple comparisons. Isolates were also categorized as pathogenic on a plant species if they caused root lesions and an average disease severity of at 5% or

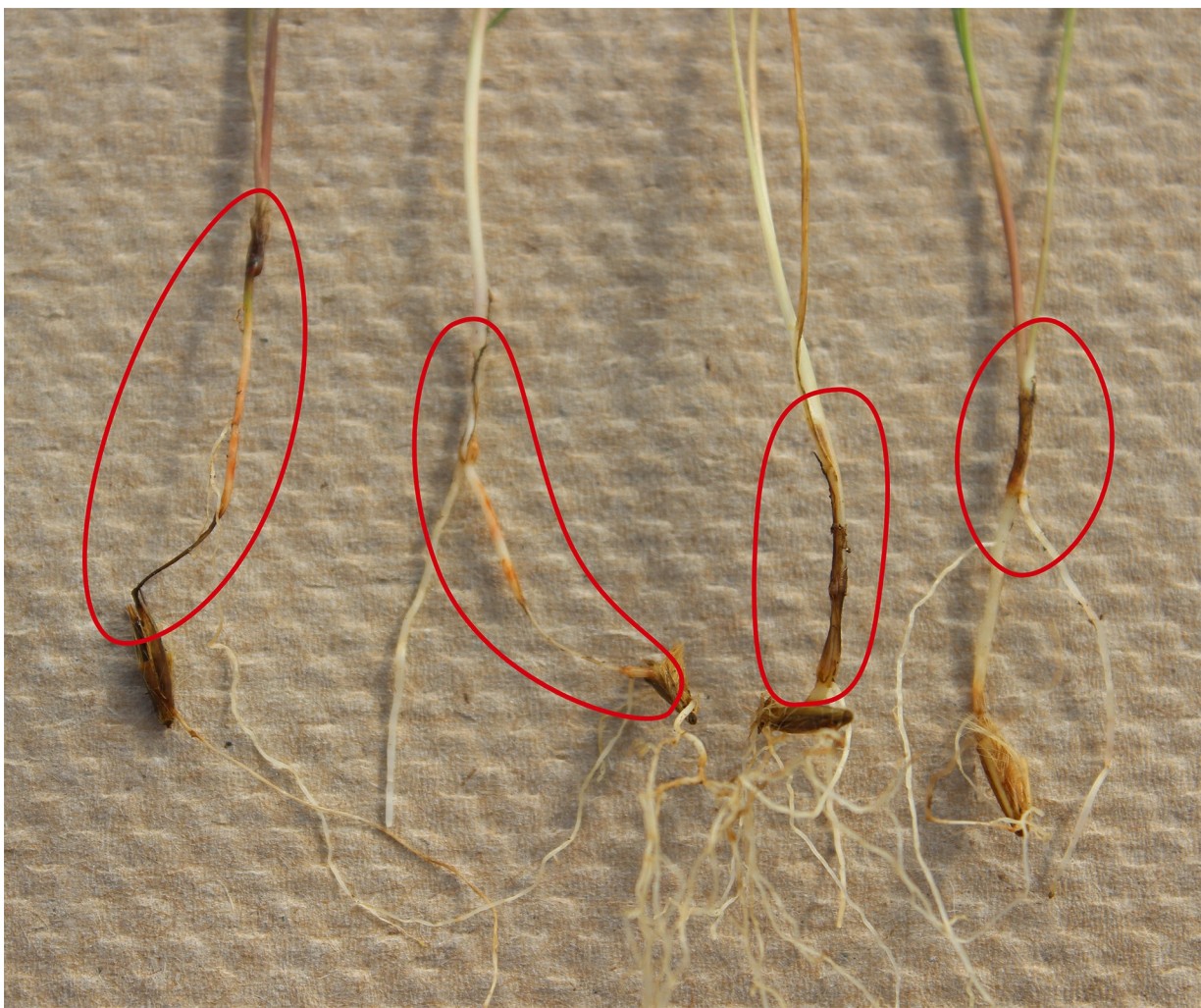

**Fig 3. The circled areas indicate necrotic lesions and reddish-brown discoloration on the sheath, crown and root of 36-day-old sand bluestem grass which had been inoculated with two agar plugs of 7-day-old binucleate *Rhizoctonia* isolate AG-B, GR2057.**

higher; this particular threshold was used because it is the lowest disease severity on the rating scale that corresponds with a disease rating of "1". Additionally, isolates causing 5 to 20% disease severity on a host were categorized as mildly aggressive on that host, 21 to 35% as aggressive, and >35% as highly aggressive.

## Results

A total of 17 isolates identified by morphological features as *Rhizoctonia* spp. were obtained from seven of 11 locations; six were collected in 2016 and 11 were isolated in 2017 (Table 1). These 17 *Rhizoctonia* isolates originated from just 15 samples in seven of the 11 locations, which is an 18% rate of isolation. An equal number of samples were collected from each county, except Logan and Hooker Counties. Samples were collected from three field locations in Logan County and two field locations in Hooker County. In terms of which type of sample yielded the most isolates, 11 of 17 came from root zone soil; however, only one *Rhizoctonia* sp. was isolated from soil collected in 2016.

The plant species that yielded the most isolates was needle-and-thread plant (n = 10). Lower numbers were obtained from sand bluestem (n = 6), and little bluestem (n = 1), while no isolates were obtained from the roots or root zone soil of prairie sandreed. Four of the six isolates in the 2016 collection were isolated from the different roots of needle-and-thread grass plants collected from two different counties and the other two were isolated from the little bluestem soil and roots of sand bluestem plant. In 2017, 10 of the 11 isolates were collected from the soil, one isolate was collected from soil of both needle-and-thread and sand bluestem that were collected from five locations, and the other one isolate, *R. zeae* was isolated from the roots of a needle-and-thread grass plant.

In terms of counties that yielded *Rhizoctonia* spp., only five of the eight counties surveyed produced *Rhizoctonia* isolates; no isolates were obtained from Arthur, Keith or Thomas Counties (Fig 1). More than half of the isolates were from Rock and Logan Counties, which were collected in both years from sand bluestem and needle-and-thread plants, plus the only isolate from little bluestem in Logan County. In Grant, Hooker, and Logan Counties, isolates were obtained only from soil and not from any of the plant roots. Isolates from Brown County were collected only from needle-and-thread plants in both years, however, two of those isolates originated from roots in the same sample (isolate IDs GR1264 and GR1265). Two isolates from Rock County also originated from roots collected in the same sample (isolate IDs GR1248 and GR1249).

The ITS and β-tub gene sequences were used to determine that more than 70% of the isolates (12 isolates, with six in each year) were binucleate *Rhizoctonia*. Five AGs were identified that belong to binucleate *Rhizoctonia* (AG-B, -C, -F, -K, and -L). Interestingly, all of the AG-F originated from plant roots and all other binucleate *Rhizoctonia* AGs originated from soil. Although five of six isolates from 2016 collections were identified as AG-F, four of these originated from samples of needle-and-thread collected in two locations: isolates GR1264 and GR1265 from roots in one sample, and GR1248 and GR1249 from roots in one sample. All other binucleate *Rhizoctonia* AGs originated from root zone soil. The only isolate associated with little bluestem was identified as binucleate *Rhizoctonia* AG-L, which came from root zone soil collected in Logan County.

Isolates identified as one of the multinucleate *Rhizoctonia* species were only obtained in 2017. Two AGs of *R. solani* were identified as AG-3 PT and AG-4 HG II, and one isolate of *R. zeae* was collected from roots of needle-and-thread plant. Thus, just two *Rhizoctonia* were directly isolated from plant roots, binucleate *Rhizoctonia* AG-F and *R. zeae*, while the other six were isolated from soil.

There was only one location (Rock County location 1) in which isolates of *Rhizoctonia* spp. were obtained in both sampling years. In 2016, three binucleate *Rhizoctonia* AG-F were isolated from needle-and-thread plant roots in one sample and sand bluestem plant roots in one sample, while in 2017, two *R. solani* AG-3 PT were isolated from root zone soil associated with needle-and-thread and sand bluestem plants.

## Pathogenicity of isolates in soybean rolled towel assay

All 17 isolates tested in the soybean rolled towel assay produced deep lesions that extended over 40% of the root and hypocotyl areas, whereas none of control seedlings exhibited any symptoms (Table 2). Therefore, all of the isolates were considered to be pathogenic on soybean.

## Aggressiveness of isolates of *Rhizoctonia* spp. on soybean and native grasses in the greenhouse

Nine isolates of *Rhizoctonia* were selected as a representative of each species or AG and evaluated in the greenhouse on soybean, sand bluestem, and needle-and-thread plants. Variation in

**Table 1. List of *Rhizoctonia* spp. isolated from seven non-cultivated sites in the Sandhills.**

| Isolate ID | Identity[a] | Genbank accessions | | County | Location | Associated grass | Source | Year |
|---|---|---|---|---|---|---|---|---|
| | | ITS | BT | | | | | |
| GR1246 | Binucleate *Rhizoctonia* AG-F | MT874890 | MT892838 | Rock | 1 | Sand bluestem | Root | 2016 |
| GR1248[b] | Binucleate *Rhizoctonia* AG-F | MT874891 | MT892839 | Rock | 1 | Needle-and-thread | Root | 2016 |
| GR1249[b] | Binucleate *Rhizoctonia* AG-F | NA* | MT892840 | Rock | 1 | Needle-and-thread | Root | 2016 |
| GR1264[c] | Binucleate *Rhizoctonia* AG-F | MT874892 | MT892841 | Brown | 1 | Needle-and-thread | Root | 2016 |
| GR1265[c] | Binucleate *Rhizoctonia* AG-F | MT874893 | NA* | Brown | 1 | Needle-and-thread | Root | 2016 |
| GR1877 | Binucleate *Rhizoctonia* AG-L | NA* | MT892842 | Logan | 2 | Little bluestem | Soil | 2016 |
| GR2003 | *Rhizoctonia zeae* | MT874894 | MT892843 | Brown | 2 | Needle-and-thread | Root | 2017 |
| GR2014 | Binucleate *Rhizoctonia* AG-K | NA* | NA* | Logan | 1 | Needle-and-thread | Soil | 2017 |
| GR2015 | Binucleate *Rhizoctonia* AG-K | MT874895 | NA* | Logan | 1 | Sand bluestem | Soil | 2017 |
| GR2016 | *Rhizoctonia solani* AG-3 PT | MT874896 | NA* | Rock | 1 | Sand bluestem | Soil | 2017 |
| GR2017 | *Rhizoctonia solani* AG-3 PT | MT874897 | NA* | Rock | 1 | Needle-and-thread | Soil | 2017 |
| GR2051 | *Rhizoctonia solani* AG-4 HG II | MT874898 | MT892844 | Logan | 2 | Sand bluestem | Soil | 2017 |
| GR2052 | *Rhizoctonia solani* AG-4 HG II | MT874899 | NA* | Logan | 2 | Needle-and-thread | Soil | 2017 |
| GR2055 | Binucleate *Rhizoctonia* AG-C | NA* | NA* | Hooker | 1 | Sand bluestem | Soil | 2017 |
| GR2056 | Binucleate *Rhizoctonia* AG-C | NA* | MT892845 | Hooker | 1 | Needle-and-thread | Soil | 2017 |
| GR2057 | Binucleate *Rhizoctonia* AG-B | MT874900 | MT892846 | Grant | 1 | Sand bluestem | Soil | 2017 |
| GR2058 | Binucleate *Rhizoctonia* AG-B | MT874901 | MT892847 | Grant | 1 | Needle-and-thread | Soil | 2017 |

[a]AG = Anastomosis group. [b,c] Isolates with the same letter originated from the same sample. Binucleate *Rhizoctonia* (teleomorph: *Ceratobasidium* spp.). *Rhizoctonia solani* (teleomorph: *Thanatephorus cucumeris*), *Rhizoctonia zeae* (teleomorph: *Waitea circinata* var. *zeae*). Associated grass species: Sand bluestem (*Andropogon hallii* Hack), little bluestem (*Schizachyrium scoparium* [Michx.]), and needle-and-thread (*Herpersotipa comata* Trin. & Rupr.).

NA* = Not applicable/available.

disease severity was observed on the three plant species among the isolates tested. Symptoms produced on the grasses were similar in appearance to the symptoms induced on soybean (Fig 3). Among the three plant species, the average disease severity across the nine isolates was highest on sand bluestem ($\mu$ = 25%, range 10–40%), followed by soybean ($\mu$ = 16%, range 2–86%) and lowest on needle-and-thread ($\mu$ = 15%, range 8–23%). Some of the non-inoculated control plants displayed superficial root lesions with less than 5% severity rating that were attributed to physiological stress of growing in artificial conditions. All nine isolates, in contrast, caused root lesions with an average disease severity $\geq$ 8% in both sand bluestem and needle-and-thread plants (Table 3). Although mean disease severity for some of the isolates was not significantly different from the control because of between-replication variability, the marked qualitative differences in symptoms between inoculated and control plants indicated that all of the fungal isolates were pathogenic on sand bluestem and needle-and-thread. On sand bluestem, two were highly aggressive (AG-B and isolate GR1264 of AG-F), four were aggressive (AG-C, GR1248 of AG-F, AG-K, and AG-3), and the remaining three were mildly aggressive. On needle-and-thread, none were highly aggressive, two were aggressive (AG-B, AG-3), and the remaining were mildly aggressive. On soybean, one isolate was highly aggressive (AG-4), one was aggressive (AG-C), four were mildly aggressive (both AG-F, AG-K, and AG-L), and three were non-pathogenic (AG-B, -3, and *R. zeae*).

Nearly all of the binucleate *Rhizoctonia* were pathogenic on both native grasses and soybean, usually with greater disease severity on sand bluestem (Table 3). For instance, isolate ID

**Table 2. Pathogenicity of *Rhizoctonia* spp. isolates from grasses assessed on soybean line CR2603 in rolled towel bioassay.** Isolates collected in 2016 and 2017 were evaluated in separate experiments.

| Collection year | Isolate ID | Percent disease severity[a] | Identity |
|---|---|---|---|
| 2016 | GR1264 | 67 ± 2.78 a | Binucleate *Rhizoctonia* AG-F |
| 2016 | GR1248 | 62 ± 3.53 a | Binucleate *Rhizoctonia* AG-F |
| 2016 | GR1249 | 62 ± 3.06 a | Binucleate *Rhizoctonia* AG-F |
| 2016 | GR1265 | 52 ± 3.31 b | Binucleate *Rhizoctonia* AG-F |
| 2016 | GR1877 | 48 ± 4.20 b | Binucleate *Rhizoctonia* AG-L |
| 2016 | GR1246 | 45 ± 4.67 b | Binucleate *Rhizoctonia* AG-F |
| 2016 | Control | 0 ± 0 c | |
| 2017 | GR2058 | 72 ± 4.98 a | Binucleate *Rhizoctonia* AG-B |
| 2017 | GR2057 | 62 ± 4.93 ab | Binucleate *Rhizoctonia* AG-B |
| 2017 | GR2016 | 61 ± 5.09 ab | *Rhizoctonia solani* AG-3 PT |
| 2017 | GR2003 | 60 ± 4.46 ab | *R. zeae* |
| 2017 | GR2055 | 59 ± 4.89 ab | Binucleate *Rhizoctonia* AG-C |
| 2017 | GR2056 | 54 ± 5.83 bc | Binucleate *Rhizoctonia* AG-C |
| 2017 | GR2052 | 54 ± 5.07 bc | *R. solani* AG-4 HG II |
| 2017 | GR2014 | 49 ± 5.37 bc | Binucleate *Rhizoctonia* AG-K |
| 2017 | GR2015 | 44 ± 5.30 c | Binucleate *Rhizoctonia* AG-K |
| 2017 | GR2017 | 41 ± 5.69 c | *R. solani* AG-3 PT |
| 2017 | GR2051 | 40 ± 8.74 c | *R. solani* AG-4 HG II |
| 2017 | Control | 0 ± 0.20 d | |

[a] Percent disease severity ± standard error (SE), values shown are means of two experiment trials with three replications each; means with the same letter within an experiment are not significantly different based on the LSD test ($\alpha = 0.05$).

GR1264 of binucleate *Rhizoctonia* AG-F was highly aggressive on sand bluestem while mildly aggressive on needle-and-thread and soybean. In contrast, *R. solani* AG-4 HG II isolate ID GR2051 was mildly aggressive on both native grass species yet highly aggressive on soybean, causing an average of 86% disease severity (range between 62–100%), which was the most severe disease measured in this study. Symptoms produced by this isolate on soybean included

**Table 3. Disease severity in greenhouse experiments on sand bluestem, needle-and-thread, and soybean following inoculation with *Rhizoctonia* spp. isolated from native Sandhills grasses.**

| Isolate ID | Identity | Disease severity[a] | | |
|---|---|---|---|---|
| | | Sand bluestem | Needle-and-thread | Soybean |
| GR2057 | Binucleate *Rhizoctonia* AG-B | 40 ± 5.77 a | 22 ± 10.9 a | 3 ± 1.44 c |
| GR2056 | Binucleate *Rhizoctonia* AG-C | 23 ± 4.33 abcd | 19 ± 3.75 ab | 23 ± 10.9 b |
| GR1264 | Binucleate *Rhizoctonia* AG-F | 36 ± 8.51 ab | 20 ± 6.12 ab | 11 ± 6.25 bc |
| GR1248 | Binucleate *Rhizoctonia* AG-F | 30 ± 11.73 abcd | 10 ± 2.89 abc | 8 ± 2.5 c |
| GR2015 | Binucleate *Rhizoctonia* AG-K | 33 ± 10.1 abc | 10 ± 2.89 abc | 5 ± 0 c |
| GR1877 | Binucleate *Rhizoctonia* AG-L | 10 ± 2.89 de | 8 ± 2.5 bc | 5 ± 0 c |
| GR2016 | *Rhizoctonia solani* AG-3 | 24 ± 8.75 abcd | 23 ± 4.33 a | 3 ± 1.44 c |
| GR2051 | *R. solani* AG-4 | 16 ± 5.15 bcde | 10 ± 2.89 abc | 86 ± 4.25 a |
| GR2003 | *R. zeae* | 13 ± 2.5 cde | 14 ± 5.91 abc | 2 ± 0 c |
| Control | | 1 ± 1.25 e | 1 ± 1.25 c | 0 ± 0 c |

[a] Percent disease severity ± standard error (SE), means with the same letter within each column are not significantly different (LSD $\alpha \leq 0.05$); disease severity is the mean disease severity of replicates of each isolate; disease severity <5% is considered non-pathogenic, 5 to 20% is mildly aggressive, 21 to 35% is aggressive, and >35% is highly aggressive.

the seedlings being barely emerged and having deep sunken lesions with a brown to reddish-brown discoloration of the roots and an absence of secondary roots.

## Discussion

In the present study, 17 isolates of *Rhizoctonia* spp. were collected from 84 samples of soil and plants from 11 field locations over two years. It was not surprising that *Rhizoctonia* spp. would be isolated at such low frequency compared to agricultural fields that typically yield approximately two isolates per five samples [37]. As suggested by McDonald and Linde [6], it is expected that pathogen population sizes would be relatively small in locations where the host/plant diversity is relatively high, such the Sandhills grassland, which may have contributed to the low frequency of isolation in this study.

The majority of isolates of *Rhizoctonia* spp. (12 isolates) obtained in this study were classified as binucleate *Rhizoctonia* AGs, while four isolates were identified in two AGs of *R. solani*. Our results are markedly different from a previous study focused on *Rhizoctonia* from row crops that showed *R. solani* isolates outnumbered isolates of binucleate *Rhizoctonia* [38]. The difference between studies may be due to binucleate *Rhizoctonia* having a different ecological role in agricultural environments than in virgin grassland ecosystems. In row crop environments, binucleate *Rhizoctonia* most likely exist primarily as soil saprophytes on dead crop tissue or root exudates. In undisturbed grassland systems, however, they more likely are adapted to be colonizers of living roots and mildly aggressive pathogens of grasses. The fact that nearly all of the binucleate *Rhizoctonia* isolates tested in this study on sand bluestem and needle-and-thread were mildly aggressive pathogens on both grasses supports their role as adapted grass pathogens. These results are similar to findings from a study on the diversity of soilborne *Fusarium* species in Konza grasslands in Kansas, which identified some unique species of *Fusarium* present in grasslands, which were different from the species found in agricultural communities [39].

Of the 17 isolates of *Rhizoctonia* spp., 16 were collected from roots and soil associated with native grasses, needle-and-thread and sand bluestem, for which there are no previous reports of *Rhizoctonia* spp. associated with these species. One explanation is that these grasses may be more susceptible to *Rhizoctonia* spp. compared to the other two grasses sampled in this study. A subset of eight isolates from needle-and-thread and sand bluestem were found to be aggressive or mildly aggressive on the grass species in the greenhouse assay, showing that these isolates were indeed pathogenic. Unfortunately, there is a paucity of information on *Rhizoctonia* and other plant pathogens on native grasses that are not also grown for ornamental purposes (such as the case with little bluestem, which has had 243 fungal pathogens reported) [23]. In fact, none of the *Rhizoctonia* spp. identified in the present study have been previously reported as a pathogen of any native grass species known in the Sandhills [18, 23]. As such, we are not able to conclude whether the inherent susceptibility of each of these plant species may be an underlying reason why these two plant species yielded the most *Rhizoctonia* in the present study. Additional isolation of *Rhizoctonia* spp. from native grasses and testing for aggressiveness on other grasses, including little bluestem and prairie sandreed, would be necessary to confirm differential susceptibility. Interestingly, although *R. oryzae* has been reported on little bluestem previously [18, 23] we did not isolate it in the present study. This, again, may be due to the fact that little bluestem is grown for ornamental purposes, so may not have acquired the pathogen from native plant soils.

Several of the *Rhizoctonia* spp. identified in this study are known to be pathogens of cultivated crops. For example, *R. zeae* has been reported on grasses grown as turf and cultivated grasses (corn, wheat, oats, etc.). Among crops currently grown in Nebraska counties where the Sandhills are located, corn is second only to hay production in terms of number of acres and geographic extent. Those counties are primarily on the fringes of the Sandhills and includes

some of the counties that were surveyed in the present study (e.g. Arthur, Brown, Keith, and Logan Counties). So, although prior cultivation of corn at the sites could have introduced *R. zeae*, there was no history of cultivation at any of the sites selected for this study. Thus, one explanation is that the *R. zeae* isolated in Brown County may have been introduced to the native grasses via corn that was grown in the vicinity of the Sandhills region and was spread via anthropogenic means on farm equipment or via short-distance spread of basidiospores, which are rarely observed. The alternative explanation is that *R. zeae* is endemic to the region and survives as a mildly aggressive pathogen of native grasses in the Sandhills. Interestingly, this species has been previously reported as a pathogen of several cultivated crops grown in Nebraska, including sugar beet (*Beta vulgaris*), soybean, and corn [23, 37]. Similarly, we found isolates of *R. solani* AG-4, which is considered an economically important seedling pathogen of soybean and several other row crops. In the greenhouse, this isolate was more aggressive and created greater disease severity on soybean compared to the two grass species. Yet another example is the isolate of *R. solani* AG-3 PT, which is known as a pathogen of potato, which was only mildly aggressive on little bluestem grass and needle-and-thread. Taken together, it is plausible that conversion of Sandhills grassland to crop cultivation will select for *R. zeae*, which is shown in our current results is capable of being a highly aggressive disease agent.

Collectively, the results of this study suggest that the *Rhizoctonia* spp. isolated in the present study from the Sandhills grasslands are native to the grassland habitat. It is presumed that new pathogens may have emerged due to more recent conversion of grasslands into agro-ecological regions. The wild AGs from the Sandhills may represent an existing source of pathogens that could affect crops grown in areas where the Sandhills are converted into agriculture land. Additionally, our finding that Sandhills isolates of binucleate *Rhizoctonia* AG-C and AG-F, and *R. solani* AG-4 were cross-pathogenic on native grasses and soybean, supports the hypothesis that grasslands may serve as a reservoir, or place of origin, for plant pathogens that potentially could cause yield-limiting diseases on cultivated crops.

All of the isolates collected from field samples in this study caused considerable disease on soybean seedling in the rolled towel assay. While we can conclude from these results that all of the isolates may have the potential to be pathogenic on soybean given abnormal, continuously wet conditions, not all of them can be expected to cause disease on soybean under normal conditions. This was supported by the finding only a few isolates inoculated onto soybean in the greenhouse experiment caused symptoms indicative of pathogenicity on soybean. Under greenhouse conditions, isolate GR2056, representing binucleate *Rhizoctonia* AG-C, showed significant disease severity on soybean, as well as the two grass species. Interestingly, binucleate *Rhizoctonia* sp. AG-C has not been reported to cause disease in soybean. One of two isolates representing binucleate *Rhizoctonia* AG-F, the most commonly isolated species in this study, caused only mild disease severity on soybean, both were aggressive when inoculated onto sand bluestem. We noticed that the disease symptoms produced by these isolates on roots and sheaths of grass seedlings were similar to root rot and sheath blight symptoms in row crops, corn, and wheat. Numerous studies have reported aggressiveness of certain AGs of binucleate *Rhizoctonia* on various economically important crops, for example AG-F causing damping off and root rot of tobacco in Argentina [40] and in Cuba [41]. These AG-F isolates were also observed to show aggressiveness on bean seedlings in Cuba [42]. Another binucleate *Rhizoctonia* group, AG-K, showed aggressiveness on strawberry in Israel [43] and Australia [44]. In addition, *Rhizoctonia* AG-K was isolated from various row crops such as dry beans [45] and corn in Nebraska [37].

To our knowledge, this is the first study on the diversity of *Rhizoctonia* spp. from Sandhills grasses and we were able to show these isolates were pathogenic to two of the grasses studied. Furthermore, it is the first to demonstrate cross-pathogenicity to a common row crop. As the cultivation of row crops in the Sandhills of the Great Plains increases due to the improved

irrigation systems, the aggressive native *Rhizoctonia* spp. may pose a threat to cultivated crops. These results are indicative of the importance of consideration of pathogen diversity in the Sandhills for the better management of the diseases of row crops. More broadly, however, knowledge from the Sandhills on the *Rhizoctonia* isolated and characterized in the current study may also be important for native plant conservation. More than 97% of the tallgrass prairie in eastern Nebraska has already been lost to agricultural and other uses [46]. The remaining grassland and the native species there are becoming increasingly important. Yet restoration of the native blowout penstemon (*Penstemon haydenii* S. Wats.) is affected by Rhizoctonia root rot, which was cited as one of the major limiting constraints for cultivation [47]. Despite this, previous studies have not characterized species of *Rhizoctonia* responsible for this disease. Thus, information on *Rhizoctonia* spp. diversity and the collection of isolates generated from the present study may be useful for improved penstemon conservation.

## Acknowledgments

We thank Rebecca Higgins for her help in formatting the figures and tables. We also thank Monsanto Company for providing the soybean seed used in this study.

## Author Contributions

**Conceptualization:** Gary Y. Yuen.

**Data curation:** Sydney E. Everhart.

**Formal analysis:** Srikanth Kodati, Sydney E. Everhart.

**Funding acquisition:** Anthony O. Adesemoye, Sydney E. Everhart.

**Investigation:** Srikanth Kodati, Anthony O. Adesemoye, Jerry D. Volesky.

**Methodology:** Srikanth Kodati, Anthony O. Adesemoye.

**Project administration:** Srikanth Kodati, Anthony O. Adesemoye, Sydney E. Everhart.

**Resources:** Jerry D. Volesky.

**Software:** Anthony O. Adesemoye.

**Supervision:** Anthony O. Adesemoye, Sydney E. Everhart.

**Validation:** Anthony O. Adesemoye, Sydney E. Everhart.

**Visualization:** Gary Y. Yuen, Sydney E. Everhart.

**Writing – original draft:** Srikanth Kodati.

**Writing – review & editing:** Gary Y. Yuen, Sydney E. Everhart.

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
