## [Decision Letter · Decision Letter 0]

3 Jul 2020

PONE-D-20-12800

Origin of agricultural plant pathogens: Diversity and pathogenicity of Rhizoctonia fungi associated with native prairie grasses in the Sandhills of Nebraska

PLOS ONE

Dear Dr. Everhart,

Thank you for submitting your manuscript to PLOS ONE. After careful consideration, we feel that it has merit but does not fully meet PLOS ONE’s publication criteria as it currently stands. Therefore, we invite you to submit a revised version of the manuscript that addresses the points raised during the review process.

I have completed the editorial review of your manuscript and a summary is appended below. The reviewers recommend reconsideration of your paper following a major corrections. 

Please pay particular attention to the comment regarding the needful corrections, when resubmitting your manuscript, please carefully consider all issues mentioned in the reviewers' comments, outline every change made point by point, and provide suitable rebuttals for any comments not addressed.

We look forward to receiving your revised manuscript.

Kind regards,

Vijai Gupta, PhD in Microbiology

Academic Editor

PLOS ONE

Additional Editor Comments:

I have completed the editorial review of your manuscript and a summary is appended below. The reviewers recommend reconsideration of your paper following a major corrections.

Please pay particular attention to the comment regarding the needful corrections, when resubmitting your manuscript, please carefully consider all issues mentioned in the reviewers' comments, outline every change made point by point, and provide suitable rebuttals for any comments not addressed.

Journal Requirements:

We note that one or more of the authors are employed by a commercial company: Terramera Inc.

2.1. Please provide an amended Funding Statement declaring this commercial affiliation, as well as a statement regarding the Role of Funders in your study. If the funding organization did not play a role in the study design, data collection and analysis, decision to publish, or preparation of the manuscript and only provided financial support in the form of authors' salaries and/or research materials, please review your statements relating to the author contributions, and ensure you have specifically and accurately indicated the role(s) that these authors had in your study. You can update author roles in the Author Contributions section of the online submission form.

2.2. Please also provide an updated Competing Interests Statement declaring this commercial affiliation along with any other relevant declarations relating to employment, consultancy, patents, products in development, or marketed products, etc. 

4. We note that Figure 1 in your submission contain map images which may be copyrighted. All PLOS content is published under the Creative Commons Attribution License (CC BY 4.0), which means that the manuscript, images, and Supporting Information files will be freely available online, and any third party is permitted to access, download, copy, distribute, and use these materials in any way, even commercially, with proper attribution. For these reasons, we cannot publish previously copyrighted maps or satellite images created using proprietary data, such as Google software (Google Maps, Street View, and Earth). For more information, see our copyright guidelines: http://journals.plos.org/plosone/s/licenses-and-copyright.

2.1.    You may seek permission from the original copyright holder of Figure 1 to publish the content specifically under the CC BY 4.0 license.

2.2.    If you are unable to obtain permission from the original copyright holder to publish these figures under the CC BY 4.0 license or if the copyright holder’s requirements are incompatible with the CC BY 4.0 license, please either i) remove the figure or ii) supply a replacement figure that complies with the CC BY 4.0 license. Please check copyright information on all replacement figures and update the figure caption with source information. If applicable, please specify in the figure caption text when a figure is similar but not identical to the original image and is therefore for illustrative purposes only.

Reviewers' comments:

Reviewer's Responses to Questions

**Comments to the Author**

1. Is the manuscript technically sound, and do the data support the conclusions?

Reviewer #1: Partly

Reviewer #2: Yes

Reviewer #3: Partly

2. Has the statistical analysis been performed appropriately and rigorously? 

Reviewer #1: Yes

Reviewer #2: Yes

Reviewer #3: No

3. Have the authors made all data underlying the findings in their manuscript fully available?

Reviewer #1: Yes

Reviewer #2: Yes

Reviewer #3: No

4. Is the manuscript presented in an intelligible fashion and written in standard English?

Reviewer #1: No

Reviewer #2: Yes

Reviewer #3: Yes

5. Review Comments to the Author

Reviewer #1: Dear authors,

The present version is, according to me, preliminar. The present form of manuscript needs a deep English edition. In addition the papers do not fit in the standards requested for publication with the limited data presented. over in the manuscript. I have found so many mistakes throughout the manuscript. I do not see a real contribution to the field, which is deduced from the highlights and abstract itself. The experimental design is key in a scientific work and there is no indication at all in your present manuscript.

As it is presented, the manuscript is far from the standards of the scientific language.

ABSTRACT

The abstract should contain the most relevant aspects of the manuscript, including the objectives , experimental system, methodology (with novelties if any), main results and conclusions (novel knowledge generated).

DISCUSSION AND ABSTRACT

The discussion section must explore the significance of the results of the work and the contribution to the plant pathogen interaction and diversity analysis field.

Reviewer #2: The overall the MS “Origin of agricultural plant pathogens: Diversity and pathogenicity of Rhizoctonia fungi associated with native prairie grasses in the Sandhills of Nebraska" is compiled well. The MS may be considered for publication after major revision.

Minor comments

• Introduction may be reduces

• Modify the line 62-64 and it should be as e.g. a negative relationship was reported

• Modify the line 139-141 (what is meaning of one objective????)

• Introduction section should be ended with what is study in present MS

• In section “Materials and methods” Plants and approximately 1 L of root zone soil were (what is meaning 1L (1 kg????)

• Sample were collected in two year why?...... why the number of sample is different in both year

• Line 195 Rhizoctonia from plants and soil samples…..

• Line 207, plate were incubate for 48 hr. Is it possible to isolates any fungi at low temperature (22C) in 48 hr.

• No need to give the complete protocol for identification and PCR, author have used modifiy protocol or used previous one, please mention

• Where is accession number of islates?

• Many sentences are unclear please check and modify in whole MS e.g. “And also, it is the first to demonstrate cross-pathogenicity to a common row crop” this sentences is started with “AND”

• In discussion section there should be recent references.

Reviewer #3: 1. I have doubts about statistical analysis in Table 2. The authors mentioned that the same letter is not significantly different, based on the LSD test (0.05). For example, how can you say that isolate GR1264 and GR1248 doesn’t have a significant difference. You have mentioned disease percentage for one isolate 67 and another 62. Do you think there is no significant difference in both strains based on their disease percentage? Please clarify clearly.

2. Another one bigger doubt, the authors have isolated 17 isolates from three different kinds of grass. But nine strains were selected to perform the greenhouse experiment. How and what basis selected nine isolates out of 17 isolates? Please explain in detail (Line 436-437).

3. I have seen the same statistical problem in Table 3. Please look at clearly and include SE along with the value in Tables 2 & 3. Try to make some statistical graphs; it will be interesting.

4. The authors have mentioned that the used grasses are susceptible to Rhizoctonia spp. as compared to other grasses in this study. Please justify clearly, or I will suggest the authors perform interaction between the grass and Rhizoctonia sp. based on a confocal fluorescence microscope. It will give clear information.

5. The page number is missing in a few of the references. Also, scientific names should be mentioned in italics in the reference section. Please correct it carefully.

6. Figure 1 & 3 is not precise. Please increase the DPI and the quality of the image.

6. PLOS authors have the option to publish the peer review history of their article (what does this mean?). If published, this will include your full peer review and any attached files.

Reviewer #1: No

Reviewer #2: **Yes: **Ajar Nath Yadav

Reviewer #3: No

---

## [Author Response · Author response to Decision Letter 0]

22 Oct 2020

Hello Dr. Gupta, 

I believe we have addressed all of your suggested changes and comments, as well as comments from the reviewers. Our response to the reviewers attachment explains our actions taken in response to all recommendations. We greatly appreciate the time that the reviewers and you have taken to read and comment on our manuscript. 

One final comment is in response to the affiliation of one of our authors with a company. That person did not work for the company, Terramera, at the time that they contributed to this manuscript. Thus, their contributions to the company would not change and there remains no influence of that company on the research conducted in the present study, it was simply that the co-author took a new job at a company after their contribution was made. Please advise us in how to proceed and if any change is needed.

Sincerely,

Sydney Everhart

---

## [Decision Letter · Decision Letter 1]

12 Mar 2021

PONE-D-20-12800R1

Origin of agricultural plant pathogens: Diversity and pathogenicity of Rhizoctonia fungi associated with native prairie grasses in the Sandhills of Nebraska

PLOS ONE

Dear Dr. Everhart,

Thank you for submitting your manuscript to PLOS ONE. After careful consideration, we feel that it has merit but does not fully meet PLOS ONE’s publication criteria as it currently stands. Therefore, we invite you to submit a revised version of the manuscript that addresses the points raised during the review process.

MS 'Origin of agricultural plant pathogens: Diversity and pathogenicity of Rhizoctonia fungi associated with native prairie grasses in the Sandhills of Nebraska' needs minor corrections.  Kindly do the needful corrections and submit a revised MS. 

We look forward to receiving your revised manuscript.

Kind regards,

Vijai Gupta, PhD in Microbiology

Academic Editor

PLOS ONE

Journal Requirements:

Additional Editor Comments (if provided):

MS 'Origin of agricultural plant pathogens: Diversity and pathogenicity of Rhizoctonia fungi associated with native prairie grasses in the Sandhills of Nebraska' needs minor corrections. Kindly do the needful corrections and submit a revised MS.

Reviewers' comments:

Reviewer's Responses to Questions

**Comments to the Author**

1. If the authors have adequately addressed your comments raised in a previous round of review and you feel that this manuscript is now acceptable for publication, you may indicate that here to bypass the “Comments to the Author” section, enter your conflict of interest statement in the “Confidential to Editor” section, and submit your "Accept" recommendation.

Reviewer #2: All comments have been addressed

Reviewer #3: All comments have been addressed

Reviewer #4: All comments have been addressed

2. Is the manuscript technically sound, and do the data support the conclusions?

Reviewer #2: Yes

Reviewer #3: Yes

Reviewer #4: Yes

3. Has the statistical analysis been performed appropriately and rigorously? 

Reviewer #2: Yes

Reviewer #3: Yes

Reviewer #4: Yes

4. Have the authors made all data underlying the findings in their manuscript fully available?

Reviewer #2: Yes

Reviewer #3: Yes

Reviewer #4: Yes

5. Is the manuscript presented in an intelligible fashion and written in standard English?

Reviewer #2: Yes

Reviewer #3: Yes

Reviewer #4: Yes

6. Review Comments to the Author

Reviewer #2: (No Response)

Reviewer #3: The authors has addressed the comments very carefully. The manuscript looks very relevant and interesting. The manuscript is possible for acceptance publication.

Reviewer #4: Dear Authors

After my careful study of this valuable manuscript. I found that this MS is interesting and revised as per comments. Now this MS can be considered after following minor changes.

Line 131-134 - kindly re-frame this sentence.

178 - Kindly mention the percentage of alcohol

Kindly abbreviate the genus name after first use in text.

Line 311-312 kindly re-write this sentence

Line 339 and 341 - Kindly check the spelling of Conetainers

7. PLOS authors have the option to publish the peer review history of their article (what does this mean?). If published, this will include your full peer review and any attached files.

Reviewer #2: **Yes: **Dr Ajar Nath Yadav

Reviewer #3: **Yes: **Ajit Kumar Passari

Reviewer #4: No

---

## [Author Response · Author response to Decision Letter 1]

13 Mar 2021

All changes were made as suggested by the reviewer. We also completed the PACE image analysis, as described.

Response to comments from Reviewer 4: 

• Line 131-134 - kindly re-frame this sentence

Reframed the sentence

• 178 - Kindly mention the percentage of alcohol

Done

• Kindly abbreviate the genus name after first use in text

Done

• Line 311-312 kindly re-write this sentence

Done

• Line 339 and 341 - Kindly check the spelling of Conetainers

Although it is possible to find the word “conetainers” used in manuscripts, most web searches show it as a hyphenated word. We have updated our manuscript to use the hyphenated version.

---

## [Editor Report · Decision Letter 2]

17 Mar 2021

Origin of agricultural plant pathogens: Diversity and pathogenicity of Rhizoctonia fungi associated with native prairie grasses in the Sandhills of Nebraska

PONE-D-20-12800R2

Dear Dr. Everhart,

We’re pleased to inform you that your manuscript has been judged scientifically suitable for publication and will be formally accepted for publication once it meets all outstanding technical requirements.

Kind regards,

Vijai Gupta, PhD in Microbiology

Academic Editor

PLOS ONE

Additional Editor Comments (optional):

All editorial, as well as reviewers comments, have been addressed. I recommend the publication of the paper in PLOS One.
---

## [Editor Report · Acceptance letter]

16 Apr 2021

PONE-D-20-12800R2 

Origin of agricultural plant pathogens: Diversity and pathogenicity of *Rhizoctonia* fungi associated with native prairie grasses in the Sandhills of Nebraska 

Dear Dr. Everhart:

I'm pleased to inform you that your manuscript has been deemed suitable for publication in PLOS ONE. Congratulations! Your manuscript is now with our production department. 

Kind regards, 

on behalf of

Dr. Vijai Gupta 

Academic Editor

PLOS ONE